# Electron–acoustic phonon coupling in single crystal CH$_3$NH$_3$PbI$_3$ perovskites revealed by coherent acoustic phonons

Pierre-Adrien Mante[1], Constantinos C. Stoumpos[2], Mercouri G. Kanatzidis[2] & Arkady Yartsev[1]

Despite the great amount of attention CH$_3$NH$_3$PbI$_3$ has received for its solar cell application, intrinsic properties of this material are still largely unknown. Mobility of charges is a quintessential property in this aspect; however, there is still no clear understanding of electron transport, as reported values span over three orders of magnitude. Here we develop a method to measure the electron and hole deformation potentials using coherent acoustic phonons generated by femtosecond laser pulses. We apply this method to characterize a CH$_3$NH$_3$PbI$_3$ single crystal. We measure the acoustic phonon properties and characterize electron-acoustic phonon scattering. Then, using the deformation potential theory, we calculate the carrier intrinsic mobility and compare it to the reported experimental and theoretical values. Our results reveal high electron and hole mobilities of 2,800 and 9,400 cm$^2$ V$^{-1}$ s$^{-1}$, respectively. Comparison with literature values of mobility demonstrates the potential role played by polarons in charge transport in CH$_3$NH$_3$PbI$_3$.

[1] Division of Chemical Physics, Department of Chemistry and NanoLund, Lund University, 221 00 Lund, Sweden. [2] Department of Chemistry, Northwestern University, Evanston, Illinois 60208, USA. Correspondence and requests for materials should be addressed to P.-A.M. (email: pierre-adrien.mante@chemphys.lu.se) or to A.Y. (email: arkady.yartsev@chemphys.lu.se).

Methyl ammonium lead iodide, $CH_3NH_3PbI_3$, is an organic-inorganic semiconductor that has received an enormous amount of attention in recent years for its potential in solar cell technology[1–4]. The efficiency of these solar cells has increased exponentially, opening the way to cheap solar cell fabrication. In addition to the solar cell capabilities, this perovskite material also has the potential to make an impact in other technologies such as thermoelectric[5] and optoelectronics[6]. Despite the huge amount of attention this material has received, its fundamental properties are still far from being well understood. It is, for example, difficult to have a clear picture of electronic properties such as mobility, which is a key property for the development of the applications we mentioned above[7–10]. Indeed, the reported electron mobilities in $CH_3NH_3PbI_3$ spread over almost three orders of magnitude[9,10]. Hall effect measurements[7] and THz spectroscopy[8,9] are two techniques that allow determination of mobility and that have been applied to single crystal $CH_3NH_3PbI_3$. However, in both cases the measured mobilities are sample dependent and thus it is difficult for other researchers to relate their work to these values. It is, therefore, critical to develop an understanding of material properties in the simplest form: the single crystal.

Within a crystal, mobility is limited by intrinsic and extrinsic factors. Extrinsic factors, such as defects, are sample dependent, while intrinsic are properties of the material. Understanding of the electron-lattice coupling and its importance for charge transport has, therefore, attracted a lot of attention in recent years[5,11–17]. Despite this intense scrutiny, a consensus concerning charge transport has still not been reached. For instance, a $T^{-3/2}$ dependence has been reported for the mobility, which is characteristic of electron–acoustic phonon scattering[11]. Furthermore, the formation of polarons due to strong interaction of charges with the essentially polar lattice has been proposed to explain this temperature dependence and the lack of defects and optical phonon scattering[12]. However, the study of the broadening of the photoluminescence as a function of temperature points towards the important role played by optical phonons[13]. To understand charge transport, it is thus crucial to address the multiple scattering mechanisms individually. Multiple theoretical studies have been devoted to obtaining the acoustic phonon limited mobility[5,16,17]. High values, up to a few thousands of $cm^2 V^{-1} s^{-1}$, have been reported for both electron and hole mobilities, however, discrepancies exist between these reports regarding the relative contributions of holes and electrons to the overall mobilities[5,17]. Furthermore, due to the lack of experimental methods, there are, to our knowledge, no experimentally determined values of these intrinsic mobilities. Knowledge of the acoustic phonon properties and electron–phonon interactions will provide a deeper understanding of the microscopic phenomena ruling electron transport in $CH_3NH_3PbI_3$.

Here we apply picosecond ultrasonics to study $CH_3NH_3PbI_3$ single crystals[18–23], and more specifically to evaluate the deformation potential parameters[22,23]. We first briefly describe the deformation potential theory that relates elastic properties and carrier-phonon interaction to mobility[24]. We then present picosecond ultrasonics measurements to generate and detect coherent longitudinal acoustic phonons (CAPs) in a single crystal of $CH_3NH_3PbI_3$. We observe the propagation of these phonons and extract their sound velocity and the elastic constants of $CH_3NH_3PbI_3$. We then investigate the generation of CAPs and extract the deformation potentials of electrons and holes in $CH_3NH_3PbI_3$. Finally, using these properties, we calculate the electron and hole mobilities. Our study reveals that, assuming effective masses, similar to literature values, carriers have high intrinsic mobility, similar to those predicted in previous first-principles calculations. Moreover, we observe a mobility three

times higher for holes than for electrons. We also compare our results to recent THz mobility measurements in $CH_3NH_3PbI_3$ single crystal[8], grown by similar methods, to reveal the influence of impurity scattering. By comparing our data to mobility values reported in the literature, we investigate the possible role played by polarons in charge transport.

## Results

**Ultrafast electron and phonon dynamics and carrier mobility.** In a material, multiple scattering processes are responsible for the reduction of carrier mobility. These scattering mechanisms can be separated in two categories: (a) intrinsic, that is, dependent on the properties of the material, like acoustic phonon scattering, and (b) extrinsic, such as scattering with crystal defects and impurities. In a perfect crystal, extrinsic scattering processes disappear and the interaction with the lattice becomes the predominant mechanism responsible for the limited carrier mobility. Using single crystal, the influence of extrinsic scattering can already be reduced, since scattering with grain boundaries is suppressed. In the case of $CH_3NH_3PbI_3$, the temperature dependence of the mobility shows that interaction with acoustic phonons becomes the predominant scattering mechanism[11]. Using deformation potential theory, we can express the mobility as follows[16,24]:

$$\mu = \frac{(8\pi)^{1/2}\hbar^4 eC}{3(m^*)^{5/2}(k_b T)^{3/2}d^2},$$ (1)

where $e$ is the electron charge, $C$ is the elastic constant along the crystallographic axis considered, $m^*$ is the effective mass of electrons or holes and $d$ is the conduction or valence band deformation potential in the direction of interest. Deformation potentials represent the coupling between electron bands and lattice vibrations. Thus, modification of the band gap as a function of the applied pressure is related to deformation potentials. The mobility obtained via the deformation potential represents the upper limit achievable in a defect free crystal, in which the mobility is limited by scattering with acoustic phonons. From equation (1), we see that, assuming knowledge of the effective mass, $m^*$, determination of the elastic constant and of the deformation potential is sufficient to obtain this ultimate mobility.

To acquire the elastic constants and deformation potential, we used the picosecond ultrasonic technique[18–23]. This technique is based on a pump-probe scheme with femtosecond laser pulses used to generate and detect high-frequency CAPs, and has been efficiently used to assess the deformation potentials[22,23]. When a laser pulse with photon energy $h\nu$ is absorbed by a material, electrons are brought to the conduction band, creating holes in the valence band, as shown in Fig. 1a.

Photoexcited carriers have excess energy that they transfer to the lattice, which increases the temperature, and, through thermal expansion, creates a thermoelastic stress, $\sigma_{TE}$, responsible for the generation of CAPs[18,21]:

$$\sigma_{TE} = -3B\beta N\frac{(h\nu - E_g)}{C_P},$$ (2)

where $B$ is the bulk modulus, $\beta$ is the linear expansion coefficient, $N$ is the density of photoexcited carriers, $E_g$ is the bandgap of the material and $C_P$ is the heat capacity. In semiconductors, CAPs are also generated through deformation potential. Electrons are responsible for the binding between atoms, and when they are excited from the valence to the conduction band, the interatomic bonding is modified (Fig. 1b), leading to a stress, $\sigma_{DP}$. The stress created through this process is proportional to the number of photo-excited carriers, $N$, and the sum of the deformation potentials of electron in the conduction band, $d_e$, and of holes in

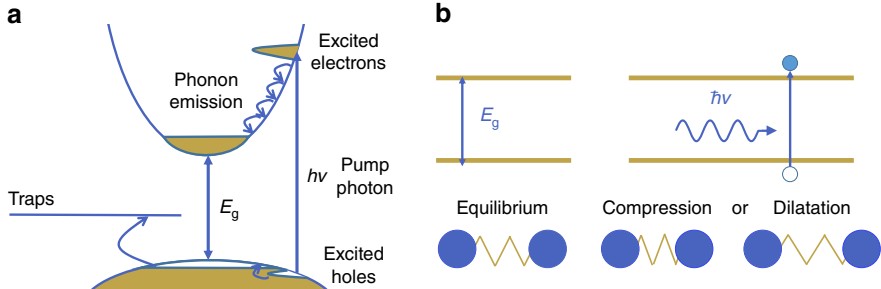

**Figure 1 | Ultrafast electron and phonon dynamics.** (**a**) Photoinduced dynamics of electrons and phonons in semiconductors. (**b**) Schema representing the generation of coherent acoustic phonons through deformation potential following the absorption of a light pulse.

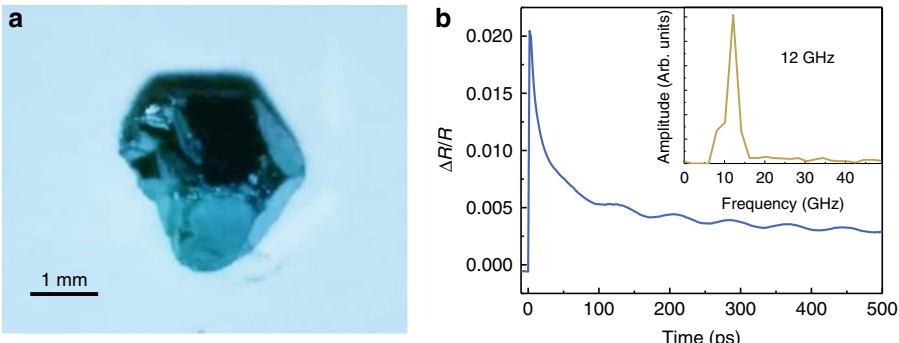

**Figure 2 | Transient reflectivity of CH$_3$NH$_3$PbI$_3$ single crystal.** (**a**) Photograph of the CH$_3$NH$_3$PbI$_3$ single crystal annealed at 100 °C. (**b**) Transient reflectivity signal obtained on the sample annealed at 100 °C at a pump wavelength of 550 nm and a probe wavelength of 850 nm. Inset: Fourier transform of the electronic background subtracted transient reflectivity.

the valence band, $d_h$ (refs 18,21):

$$\sigma_{DP} = -N(d_e + d_h), \qquad (3)$$

Generated CAPs then propagate through the material at a velocity $= \sqrt{\frac{C}{\rho}}$, with $C$, the elastic constant along the direction of propagation of the phonons and $\rho$, the mass density. Other additional mechanisms can have a role in the generation of CAPs, such as the inverse piezoelectric effect[25]. However, in the case of CH$_3$NH$_3$PbI$_3$, the piezoelectric coefficient e$_{33}$ is relatively weak with reported value of 0.07 C m$^{-2}$ (ref. 26), thus we can neglect the piezoelectric contribution to the generation of CAPs.

However, in a crystal exhibiting a large amount of defects, the picture can be different. As depicted in Fig. 1a, carriers can also relax to trap levels. As a consequence, trapped carriers dissipate a different amount of energy to the lattice, since the trap energy level is often different from the bottom of the conduction band or the top of the valence band. Additionally, if the carriers are trapped on a time scale faster than the phonon build up time, they also contribute differently to the deformation potential stress. While carriers in bands are delocalized, trapped carriers are strongly localized, and will thus create local deformations. The amplitude and rate of this deformation also depends on the nature of the traps. If a trap is due to a vacancy, the deformation will be different than the one due to an interstitial or substitutional atom. Finally, despite the scarcity of literature on the deformation potential of trap states, there are few examples showing that these deformation potentials are negligible compared with conduction and valence bands deformation potentials[21,27]. Taken that into account, we can rewrite the stress generated by the absorption of a laser pulse as follows,

considering holes as the dominantly trapped species:

$$\sigma = \sigma_{TE} + \sigma_{DP}$$
$$= -3B\beta N \frac{hv - E_g}{C_p} - 3B\beta n \frac{E_g - E_t}{C_p} - Nd_e - (N-n)d_h. \qquad (4)$$

For a given material, the amplitude of the stress depends on the density of photoexcited carriers, $N$, the excess energy, $hv - E_g$ and the trap density, $n$, and energy, $E_t$.

Here we use equation (4) to extract the deformation potentials. The method is described in details in the Supplementary Notes 1 and 2 and Supplementary Fig. 1, and can be understood as follows: first, by changing the photon energy, we modify the relative contributions of thermoelasticity and deformation potential to the overall stress. This step allows to separately obtain the thermoelastic and deformation potential contributions. Then by varying the trap density in two different samples, we modify the relative contribution of hole and electrons to the deformation potential stress.

**Pump-probe spectroscopy on CH$_3$NH$_3$PbI$_3$ single crystals.** For the experiments, two CH$_3$NH$_3$PbI$_3$ perovskite single crystals were grown (Fig. 2a)[7]. One of the samples was thermally annealed at a temperature of 100 °C, while the second was annealed at 200 °C. Using different annealing temperature, we are able to modify the density of trap within the crystal, as annealing induce evaporation of CH$_3$NH$_3$+, and formation of hole traps[28].

As we have seen, in picosecond ultrasonics, the absorption of a laser pulse, the pump, leads to generation of CAPs[18–23]. Detection of the pump-induced phenomena is done by monitoring the reflectivity of the crystal surface by a second time-delayed laser pulse, the probe. Figure 2b shows the transient reflectivity

obtained on the sample annealed at 100 °C with a pump wavelength of 550 nm and a probe wavelength of 850 nm. On a short time scale, we observe a sudden rise of the reflectivity induced by the photoexcitation of electron-hole pairs. This reflectivity then decays as carriers transfer their energy to the lattice. Additionally to that, we observe oscillations due to the propagation of CAPs. In picosecond ultrasonics, propagating CAPs are detected through Brillouin oscillations[18,19]. The phonons induce a local modification of the refractive index and the probe is then partially reflected by the CAPs as well as the surface of the sample. These two reflections interfere and the state of interference is changing with the pump-probe time delay as the difference of optical paths between the two reflections changes when the phonons propagate away from the surface. The frequency of such oscillations is given by $f = 2nv/\lambda$, with $n$, the refractive index at the probe wavelength, $\lambda$, and $v$, the sound velocity[18,19]. The inset of Fig. 2b shows the Fourier transform of the transient reflectivity after removal of the electronic contribution and reveals the frequency of the Brillouin oscillation at 12 GHz. Using a refractive index, $n$, of 2.3 (ref. 29), we find a sound velocity of 2,200 m s$^{-1}$ in good agreement with the literature value of 2,135 m s$^{-1}$ (ref. 30). Using this sound velocity, and a mass density of 4.160 g cm$^{-3}$ (ref. 7), we finally obtain the elastic constant along the direction of propagation of the phonons, $C_{11} = 20.4$ GPa. We realize that these values correspond to the $<100>$ crystallographic direction of the tetragonal $I4cm$ space group which coincide with the hexagonal facets of the $CH_3NH_3PbI_3$ single-crystal, in good agreement with previous measurements[31].

To obtain the mobility of charges, we need to determine the deformation potential in addition to the elastic constants. As mentioned previously, by changing the pump wavelength, we can modify the relative contribution of the deformation potential and thermoelastic stress to the overall stress as the amount of dissipated heat is determined by the difference between the excitation photon energy and the band gap, as we showed in equation (2). According to Hooke's law, the amplitude of the generated CAPs is proportional to the stress[32]. Therefore, we can extract the deformation potential induced stress by comparing the amplitudes of the generated CAPs for different pump wavelengths (Supplementary Notes 1 and 2, and Supplementary Fig. 1). However, the measured amplitude is only representative of a part of the whole acoustic strain, since we are measuring phonons at the specific Brillouin frequency. Furthermore, different pump wavelengths have different penetration depth, leading to different frequency spectra of the generated CAPs. We must thus introduce a correction factor to take into account the change of the frequency content of the CAPs as a function of the pump wavelength (Supplementary Note 1 and Supplementary Fig. 1). In Fig. 3, we show experiments with pump wavelengths of 550 and 650 nm and a probe wavelength of 850 nm.

We observe a variation of the amplitude of the generated CAPs due to the modification of the relative contribution of the thermoelastic and the deformation potential stress. Performing a linear combination between the stresses (equation (4)) for each photon energy, we obtain the deformation potential induced stress, corresponding to the two last terms of equation (4). Assuming $n << N$, we can estimate the band gap deformation potential, that is, the sum of the deformation potential of the valence and conduction bands, $d_e + d_h = -3.93$ eV which is in agreement with values reported in the literature of $-4.74$ (ref. 16), and $-2.5$ eV (ref. 33). To confirm our method, we used a GaAs substrate as a benchmark material, and performed similar measurements. We found a band gap deformation potential of $-9.5$ eV, very close to the literature values that are between $-9$

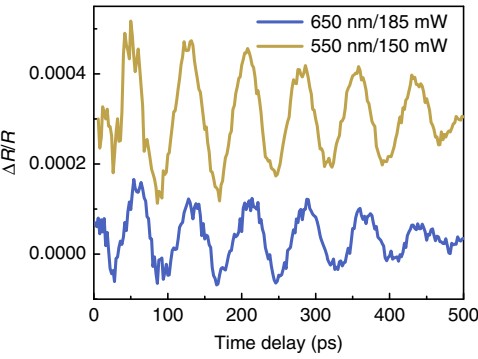

**Figure 3 | Wavelength dependent strain generation. (a)** Transient reflectivity signals at a constant probe wavelength of 850 nm, and a pump wavelength of either 550 or 650 nm of the $CH_3NH_3PbI_3$ single crystal annealed at 100 °C. The difference in amplitude is due to the change of the thermoelastic and deformation potential contributions with changing pump wavelength.

and $-10$ eV (ref. 21) (Supplementary Note 3 and Supplementary Figs 2 and 3).

The determined bandgap deformation potential includes the contributions of holes in the valence band and of electrons in the conduction band. Therefore, we cannot yet estimate the electron and hole mobility separately. To distinguish the valence and conduction band deformation potentials is a non-trivial task as obviously electrons and holes are generated simultaneously. Nevertheless, in equation (4), we see that imperfections create an asymmetry in the role played by holes and electrons on the stress. It thus becomes possible to extract separately the deformation potentials because our two samples have different number of traps (Supplementary Note 2). However, we need to know the energy of the trap level, $E_t$, and the density of traps, $n$, in the samples.

To obtain $E_t$, we measured the reflectivity spectra of the perovskite single crystal 10 ps after the pump pulse as depicted in Fig. 4a. We chose a value of 10 ps, which is larger than the trapping time in $CH_3NH_3PbI_3$ with high number of defects, but shorter than the recombination time of these traps[34].

We fitted the spectra using two Gaussians, corresponding to different energy levels. We observe a first contribution (light blue curve) with a lower energy level located around 765 nm, which corresponds to 1.63 eV, in agreement with the band gap of $CH_3NH_3PbI_3$ (ref. 33). The second contribution (green curve) is located below band gap, and we attribute it to trap states[35]. Thus, we obtained the energy level of the trap states, $E_t = 1.55$ eV.

To estimate the relative difference in the trap density, we extracted the CAPs decay by performing transient reflectivity experiments in both samples, as shown in Fig. 4b,c. The photon energy of the probe was chosen below the band gap of the single crystal, and thus the decay of the observed oscillations is solely due to the acoustic attenuation and not due to the probe light absorption, since traps have a low absorption cross section[36]. When comparing these two traces, we observe that in the sample annealed at 200 °C, the Brillouin oscillations are decaying faster than in the sample annealed at 100 °C. We attribute this difference to an increased number of defects in sample annealed at 200 °C. Indeed, at this phonon frequency (12 GHz) and temperature, phonon scattering with impurities is the dominant scattering mechanism. We thus assume that only defect scattering contributes to the observed decay and that it is directly proportional to the density of defects[37]. Annealing of $CH_3NH_3PbI_3$ leads to the evaporation of $CH_3NH_3^+$, and thus create defects that will scatter with acoustic phonons.

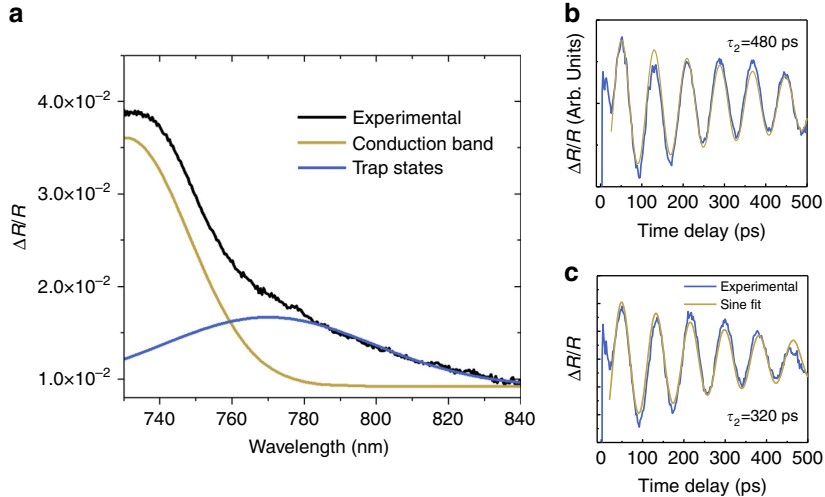

**Figure 4 | Influence of trap states. (a)** Transient reflectivity spectra obtained at a 10 ps delay (black line) and fitted spectra (red line). Two contributions are observed, from the conduction band (light blue line) and from the trap states (green line). **(b,c)** Transient reflectivity at a pump wavelength of 550 nm and a probe wavelength of 850 nm in the samples annealed at 100 and 200 °C, respectively. The difference in the decay of the phonon oscillations is the result of the stronger phonon scattering due to the defects in the sample annealed at 200 °C.

| Table 1 \| Mobility properties of $CH_3NH_3PbI_3$. | | | |
|---|---|---|---|
| | **This work** | **Experimental** | **Theory** |
| $d_e$ (eV) | − 2.93 ( ± 0.5) | | − 1.48 (ref. 16), 1.1 (ref. 17) |
| $d_h$ (eV) | − 2.2 ( ± 0.25) | | − 3.26 (ref. 16), 1.5 (ref. 17) |
| $n$ (cm$^{-3}$) | $4 \times 10^{18}$ | | |
| $\mu_e$ (cm$^2$ V$^{-1}$ s$^{-1}$) | 2,800 ( ± 950) | 2.5 (ref. 44), 24 (ref. 40), 27 (ref. 11), 66 (ref. 7), 800 (ref. 8) | 23,400 (ref. 16), 2,554 (ref. 13) |
| $\mu_h$ (cm$^2$ V$^{-1}$ s$^{-1}$) | 9,400 ( ± 3,000) | 164 (ref. 40) | 2,500 (ref. 16), 7,176 (ref. 13) |
| $C_{11}$ (GPa) | 21 | 25 (ref. 45) | 20.1 (ref. 21) |

Conduction and valence band deformation potential, trap density in sample 1, electron and hole mobility, and elastic constant measured using picosecond ultrasonics compared with experimental measurements of $CH_3NH_3PbI_3$ single crystal and theoretical predictions of deformation potentials, mobilities and elastic constant.

Furthermore, the evaporation of $CH_3NH_3^+$ results in the formation of hole traps[28], and we can, therefore, relate the acoustic phonon lifetime to the density of traps. From the analysis of the decay time in both samples, we conclude that, at a given number of photo-excited carriers, there will be 1.5 times more trapped holes in the sample annealed at 200 °C than in the one annealed at 100 °C. The observation of a higher trap density for higher annealing temperature may seem surprising, but this stems directly from the chosen method of annealing. Recent studies have shown that thermal annealing leads to the decomposition of $CH_3NH_3PbI_3$ and the evaporation of $CH_3NH_3^+$, thus increasing the defect density and hole trapping[28].

## Discussion

We have now gathered all the necessary information and can calculate the deformation potentials, trap densities and mobilities as shown in Table 1 using effective masses of 0.24 $m_e$ and 0.31 $m_e$ for electron and holes[16], respectively, with $m_e$, the electron mass (details of the calculation can be found in Supplementary Note 2).

We obtain a high acoustic phonon limited mobility for both electrons and holes. For comparison, in GaAs using room temperature value of − 8.6 eV for the conduction band deformation potential[38], 0.067 for the effective mass[38], and 118 GPa for the elastic constant[39], we obtain an acoustic phonon-limited mobility of 85,000 cm$^2$ V$^{-1}$ s$^{-1}$. The values of mobility we have obtained are in good agreement with multiple recent first-principles calculations that have predicted acoustic phonon

limited mobility of few thousands cm$^2$ V$^{-1}$ s$^{-1}$ (refs 5,16,17). This high mobility is due to the relatively weak interaction between carriers and phonons and to the small effective masses in $CH_3NH_3PbI_3$. We also notice that hole mobility is about three times larger than electron mobility, similar trend to what have recently been observed for the overall mobility of $CH_3NH_3PbI_3$ single crystals[40]. The relative importance of hole and electron mobility in $CH_3NH_3PbI_3$ has been under debate as multiple studies have reported diverging point of view[5,17]. Here we experimentally observe that electrons are more strongly scattered by lattice vibrations than holes.

If we now compare our results with other reported experimental mobilities, we observe a strong discrepancy. For example, for measurements by Hall effect in single crystal[40], the difference is of two orders. More recently, measurements done using THz spectroscopy revealed a much higher electron mobility, 800 cm$^2$ V$^{-1}$ s$^{-1}$ (ref. 8). This mobility was obtained on single crystals grown with the exact same method as ours. It is thus possible to do a meaningful comparison between these results. In our work, the mobility we observe is limited by acoustic phonon scattering and as such represents the upper limit reachable in the case of a perfect crystal. However, as we have demonstrated, the density of defects is relatively high in these samples. For such high density of defects, impurity scattering becomes the predominant mechanism ruling the mobility[17]. The lower value observed with THz spectroscopy in ref. 8 can thus be explained by the high density of defects present in the crystal that diminishes the observed mobility.

However, other reports have demonstrated a temperature dependence characteristic of acoustic phonon limited mobility in samples with a much lower mobility compared with our results[11,41–43]. Furthermore, other report suggests that acoustic phonon scattering may be negligible and that the optical phonon scattering is the dominant mechanism[13]. To unify the discussed models we propose to consider the concept of polarons[12]. It has been suggested that a strong long range Coulomb potential in $CH_3NH_3PbI_3$, could lead to formation of large polarons that would be protected from optical phonons and defect scattering[12]. The formation of a polaron can thus explain the experimentally observed temperature dependence of the mobility[11,41–43]. Moreover, polarons have a much higher effective mass than charges. Using a mobility ranging from 10 to 100 $cm^2 V^{-1} s^{-1}$ as a reference for a perfect single crystal and the deformation potential determined in this study, we calculate a polaron effective mass of 0.9–2.3 $m_e$. Additionally, the concept of polaron may also explain the lack of acoustic phonon contribution in temperature dependent photoluminescence broadening[13]. In these experiments, the broadening presents an onset that is attributed to the scattering of charges by optical phonons, which only contributes for a thermal energy large enough. However, similar onset should be observed for the scattering of polarons by acoustic phonons, as the formation of a polaron requires an activation energy. Finally, formation of a polaron will also require some time, which could explain the fast decay of photo-conductivity on a picosecond time scale from the high early value observed in THz experiments[8]. Thus a concept of polaron provides a unifying picture, but further studies are needed to confirm its exact role.

In conclusion, we have developed a new method based on the generation and detection of coherent acoustic phonons to investigate charge–acoustic phonon coupling and retrieve the acoustic phonon limited electron and hole mobilities, assuming the effective masses. We have then applied this method to a single crystal of $CH_3NH_3PbI_3$. We have measured the longitudinal sound velocity of $CH_3NH_3PbI_3$, by monitoring the propagation of travelling coherent acoustic phonons. Then we have determined the deformation potential, which represents the interaction between carriers and acoustic phonons, by monitoring the amplitude of the generated coherent acoustic phonons as a function of the excitation excess energy and defects. Finally, we have applied deformation potential theory to obtain the intrinsic mobilities of electrons and holes. We have obtained relatively high ultimate charge mobilities that can serve as a target in future $CH_3NH_3PbI_3$ device preparations. Additionally, we have compared our data with experimentally reported values of electron mobility and propose a polaron model to unify the picture of charge transport in $CH_3NH_3PbI_3$.

## Methods

**Femtosecond pump-probe spectroscopy.** Experiments were performed using a regeneratively amplified, mode-locked Yb:KGW (Ytterbium-doped potassium gadolinium tungstate) based femtosecond laser system (Pharos, Light conversion) operating at 1,030 nm and delivering pulses of 200 fs at 2 kHz repetition rate. This laser is then used to pump two non-collinearly phase-matched optical parametric amplifiers (NOPAs). A first one (Orpheus-N, Light Conversion), was used to generate pump pulses centered at 550 or 650 nm with pulse duration of about 35 fs. The second NOPA (Orpheus-N, Light Conversion), generated probe pulses at 850 nm with 40 fs pulse duration, that were time delayed with respect to the pump. The pump beam was chopped at the frequency of 1 kHz using a mechanical chopper. Both beams were focused on the sample and the modifications of the probe reflectivity induced by the pump was time-resolved.

**Synthesis of $CH_3NH_3PbI_3$ single-crystals.** Single crystals where prepared in a similar manner to the one previously reported[8]. Briefly, single-crystals of $CH_3NH_3PbI_3$ suitable for characterization were grown from a solvent mixture comprising of aqueous HI (57% w/w, 5.1 ml) and aqueous $H_3PO_2$ (50% w/w,

1.7 ml) placed in a 20 ml scintillation vial. In the acid mixture, PbO (670 mg, 3 mmol) and $CH_3NH_3Cl$ (202 mg, 3 mmol) were added sequentially leading to the formation of a black precipitate, which rapidly dissolved as the temperature was raised to the boiling point of the mixture ($\sim 130\,^\circ C$) on a hotplate. Upon complete dissolution, the resulting clear, bright yellow solution was removed from the hotplate and the vial was covered while hot. On standing, upon reaching ambient temperature a small number of crystals begin to form, which act as initial seeds for the subsequent crystal growth from the supersaturated supernatant solution. Well-formed, faceted crystals of rhombic dodecahedral crystal habit were obtained after 2 weeks. The crystals were collected manually by decanting the mother liquor, pressed dry with a soft filtration paper and thoroughly dried under a stream of $N_2$ gas.

**Annealing of the $CH_3NH_3PbI_3$ single-crystals.** The obtained crystals were manually selected and transferred into 9 mm OD fused five silica tubes (roughly five crystals per tube). The tubes were evacuated to $10^{-3}$ mbar and flame sealed. The tubes were immersed into a sand bath standing at 100 or 200 $^\circ C$, respectively, so that approximately 2/3 of the tubes remained outside the sand level. The tubes were maintained to the said temperatures for 2 h followed by air quenching to ambient temperature. This method strictly requires freshly isolated single-crystals in order to prevent damage to the surface of the crystals. When the crystals are fresh, the crystals retain their original luster after the annealing process and no degradation is observed. However, when the isolated crystals have been exposed to the atmosphere for a long period of time (about 4 weeks in this work) before annealing, then the annealing process has a severe effect on the crystals, leading to surface degradation forming a yellow crust, presumably due to formation of $PbI_2$. The coverage of the yellow crust scales with annealing temperature, so that at 100 $^\circ C$ the coverage is partial whereas at 200 $^\circ C$ the coverage is nearly complete by visual inspection. For the purposes of the present study, only the latter two samples were characterized aiming to systematically vary the defect concentration.

**Data availability.** The authors declare that the data supporting the findings of this study are available from the corresponding authors upon request.

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

## Acknowledgements

We would like to thank Dr Eva Unger, Dr Kaibo Zheng and Dr Carlito S. Ponseca for fruitful discussions. The work at Lund University was supported by the Crafoord Foundation and the Knut and Alice Wallenberg Foundation. The work at Northwestern University was supported by grant SC0012541 from the U.S. Department of Energy, Office of Science.

## Authors contributions

P.-A.M. developed the model, conceived and performed the experiments. C.C.S. and M.G.K. prepared the samples. P.-A.M and A.Y. analysed the data, wrote the manuscript and all authors approved the final version.

## Additional information

**Competing financial interests:** The authors declare no competing financial interests.

**Publisher's note**: 

