## [Peer Review File · Nature Communications]

Reviewers' Comments:

Reviewer #1 (Remarks to the Author)

Report on the manuscript entitled: " High Charge Mobility Limit in Single Crystal CH₃NH₃PbI₃ Perovskites Revealed by Coherent Acoustic Phonons » by Mante et al.

The authors propose to employ a time-resolved optical method to evaluate the electron-hole-acoustic phonon coupling strength, namely, the electron-hole-acoustic phonon deformation potential in the CH₃NH₃PbI₃ perovskites. For this purpose, the authors make use of femtosecond laser to generate and to detect coherent acoustic phonons. In order to separate the contributions from the thermoelastic and the deformation potential term, the authors perform a multiwavelength investigation. Based on their evaluation of the electron-hole acoustic phonon deformation potential value, they then evaluate the carriers (electrons, holes) mobility which is an important parameter for the photovoltaic properties of organic perovskites. The idea to use femtosecond laser to discuss the electron-hole-acoustic phonon deformation potential is interesting even not completely new (see comments). But it is worth to go on with this technique and to try to apply it to the organic perovskites.

The paper is well written but does not sound fully technically correct since the analysis is based on a wrong theoretical statement despite the theory is already well established in the literature and verified by experimental works. As a matter of fact, my opinion is that the claims are not enough supported and sometimes speculative unfortunately. I think the paper does not meet the criteria of Nature Communications in this form.

Comments :

1) As far as I know, the study of coherent acoustic phonon generation with femtosecond laser has already been employed to try to evaluate/confirm the deformation potential parameter (electronic pressure) in metals (Perner et al, Phys. Rev. Lett. 85 (2000) and in perovskite structure RNiO₃ (Ruello et al Phys. Rev. B 79, 094303 (2009)). The authors could mention these references which, in the case of publication, would provide to the broad audience of Nature Communications a confirmation of the potentialities of the technique.

2) The authors develop an analysis based on the measured amplitude of coherent acoustic phonon signals and relate (see SI) these values to the electron-hole-acoustic phonon deformation potential. In the present manuscript, there is the following statement

"We know that, for the same photoexcited carrier density, the difference in amplitude for different pump wavelengths is solely due to the change of energy of the pump photons »

This statement is not true in general. The authors have to support this approximation with more serious arguments. As the authors certainly know, the amplitude of the CAP does depend on the detection and generation processes. The detection is performed at the Brillouin frequency. So the amplitude of the photoinduced strain does not depend only on the values of the deformation potential, the thermal expansion, the bulk modulus, heat capacity. The detected amplitude does depend on the amplitude of the photoinduced strain spectrum at the Brillouin frequency. It is established for more than 20 years that the photoinduced strain (in the simplest form) has to be written as a function of the acoustic phonons frequency ω and of the physical parameters of generation (see well established theory and experimental confirmations in Gusev & Karabutov Laser optoacoustics, AIP, New York 1995. Wright et al, Phys. Rev. 64, 081202 (2001), Babilotte et al Phys. Rev. B 81, 245207 (2010), Young et al Phys. Rev. B, 86, 155207 (2012)):

As an example that the authors know, for a bipolar strain pulse (typically the one the authors might generate in their experiments), this spectrum $S(\omega)$ can be written as :

$$S(\omega) \propto \omega \tau / (1 + \omega^2 \tau^2)$$

where τ is a characteristic time that can be associated to different physical parameters during the generation process. In the case the spatial extension of the pulse strain is only limited by the skin depth of pump penetration (that depends on the pump wavelength), τ corresponds to the time propagation of the acoustic phonon within the skin of pump light. And τ is a function of the pump wavelength in general. This spectrum expression can be developed also in the case of diffusion of hot carriers and of heat, see references above.

- The authors have to correct first this wrong statement.

- Secondly, the consequence of this is that, without taking into account the photogenerated acoustic pulse spectrum, the authors cannot apply their model straightforwardly as they do. As a consequence, they cannot claim it is straightforward to evaluate the deformation potential parameter and the carriers mobility. The authors should present/develop a calculation of the photoinduced strain taking into account (at least) the effect of the pump optical absorption coefficient (which depends on the pump wavelength). The authors know certainly the optical properties of this perovskite.

Maybe this rapid calculation will show that their approximation appears to be right, but they do not show in the current manuscript that all the necessary checks have been taken. The readers have to be convinced first.

The authors provide in the SI a calculation about GaAs that they consider as a proof of their method. I do not believe it. GaAs has been studied by many many authors (Japan, France, UK, US, Taiwan, etc...) and all know, like the authors, that the amplitude at the Brillouin frequency does not depend only on the deformation potential (see Gusev & Karabutov *Laser optoacoustics*, AIP, New York 1995. Wright et al, *Phys. Rev.* 64, 081202 (2001), Babilotte et al *Phys. Rev. B* 81, 245207 (2010), Young et al *Phys. Rev. B*, 86, 155207 (2012)). I think the example of GaAs is not convincing.

I think that the oversimplified model used by the authors may explain why they observe a so large discrepancy in the carriers mobility with other tabulated data (see comparison in the Table 1).

3) The experimental Brillouin signals the authors report has a unusual time-dependence variation, in a sense that the Brillouin amplitude exhibits a kind of beating (appearance of a maximum in its amplitude at a time delay 100-150ps). The authors do not comment this effect. How can they fit then the Brillouin magnitude and extract the maximum of amplitude without explaining the beating effect ? This is impossible to my best knowledge.

I think the authors should take care of not concluding too rapidly without discussing at least different others possible effect (known in the literature). I mean that this kind of beating could have different origin :

- optical anisotropy of the perovskite which could lead to the detection of different component of Brillouin frequencies (ordinary, extraordinary), and this in particular in the tetragonal phase $\langle 100 \rangle$ propagation direction (see such phenomena in another perovskite BFO Lejman et al *Nature Comm* 7, 12345 (2016))

- inhomogeneities in the crystal that could lead to non-monotonous Brillouin signals (see A. Steigerwald, *Appl. Phys. Lett.* 94, 111910 (2009)).

4) Finally, it is difficult to follow the discussion about the trap dynamics and it appears somehow speculative at this stage. Why choosing the time 10ps to and to deduce so fundamental properties like traps energy ??

To my opinion, the paper does not meet the criteria of Nature Communications and cannot be published without more developed theoretical analysis and more supported analysis/discussions.

Reviewer #2 (Remarks to the Author)

The manuscript describes a new method to measure the deformation potentials of the CH₃NH₃PbI₃ single crystal, by using a pump-probe scheme that generates and detects high frequency coherent acoustic phonons (CAPs). The measured deformation potentials are compared with previous theoretical predictions, and then combined with the deformation potential theory to estimate the electron and hole mobilities due to scattering by acoustic phonons, which are supposed to set the upper limit for the charge carrier mobilities in these systems. As various properties of charge carrier transport in CH₃NH₃PbI₃ are still not very clear, this work is timely and provides important experimental data to understand charge carrier transport. However, several important problems need to be addressed before the paper can be accepted for publication in Nat. Comm.

1. The major problem concerns how the experimental results are presented. The author stated in the Abstract and a few other places that "...measure the acoustic phonon-limited electron and hole mobility using coherent acoustic phonons generated by femtosecond laser pulses". However, the mobility is not measured directly in this work, and the central quantities of the experiment should be the deformation potentials that are subsequently used to estimate the charge carrier mobility. I believe that measurement of the deformation potentials is already an important result, and the authors need to revise related parts of the paper to focus on what is really studied in the experiments.

2. In this work, the pump-probe measurement is carefully designed to obtain the deformation potentials. Even though, some important details and/or discussions are not included in the paper. For example, are all calculations involving the $(\Delta R)/R$ amplitudes compensated for the different absorption coefficients at different wavelengths? (a figure of the absorption spectra in the supplementary may also help) As a result of different absorption coefficients, do they have different penetration depths? And will this affect the amplitude of CAP signals?

3. On how the impurity densities are determined. (1) It seems that, the authors measure the impurity scattering of phonons and assume that scattering of charge carriers is proportional to that of phonons. Is this an assumption or can it be justified? If it is an assumption, it should be stated explicitly in the paper. (2) Are the absorption spectra the same for the two samples annealed at different temperatures? If not, will this also affect the time decay of the signals in Fig. 4? (It is claimed in Ref. 14 that, "the attenuation of the reflectivity oscillations is principally due to the decay of the sensitivity function.", thus the decay should also depend on the penetration depth.)

4. How accurate are the deformation potentials obtained from the experiment? And how are the error bars in the electron and hole mobilities estimated in Table I?

Reviewer #3 (Remarks to the Author)

Review of:

"High Charge Mobility Limit in Single Crystal CH₃NH₃PbI₃ Perovskites Revealed by Coherent Acoustic Phonons"

The authors present a clever measurement of the deformation potential parameters in MAPbI₃ using ultrafast laser pulses to excite coherent phonon modes and measuring the transient reflectance dynamics just below the band edge.

While I find this to be an interesting study, I am highly skeptical of the conclusions the authors draw because they are inconsistent with available experimental evidence. Using the deformation potential parameters they measure, the authors calculate the electron and hole mobility and state that this is the intrinsic upper limit for mobility in this material. They state that the mobility that is measured experimentally must be limited by extrinsic defects. This seems like a dubious conclusion for two reasons.

First, studies of the temperature-dependence of the mobility in MAPbI₃ has generally found a $T^{-3/2}$ dependence (J. Phys. Chem. Lett. 2015, 6 (24), 4991–4996.). This looks like a mobility limited by acoustic phonon scattering, consistent with the author's view. The problem is that the mobility measured in these same experiments is orders of magnitude lower than that calculated by the present authors. If the mobility were limited by defect scattering in real samples (as these authors claim) mobility should go UP with increasing temperature, not down in the experiment. They must address this disconnect.

Second, the assertion that their calculated mobility represents the intrinsic upper limit assumes that there are no other intrinsic mechanisms that could further limit the mobility. This might be true in a non-polar semiconductor, however the strongly polar, even ionic, nature of MAPbI₃ lattice suggests that the Frölich interaction should be important, leading to carrier scattering that is dominated by scattering from optical phonons as is observed in other polar semiconductors like GaAs. A recent paper by the Herz group argues strongly that this is the case. (DOI: 10.1038/ncomms11755).

Thus, I feel that major revision is in order to better place the current work into context with existing literature.

Point by point response to reviewers' comments

Reviewer 1:

1) As far as I know, the study of coherent acoustic phonon generation with femtosecond laser has already been employed to try to evaluate/confirm the deformation potential parameter (electronic pressure) in metals (Perner et al, Phys. Rev. Lett. 85 (2000) and in perovskite structure RNiO₃ (Ruello et al Phys. Rev. B 79, 094303 (2009)). The authors could mention these references which, in the case of publication, would provide to the broad audience of Nature Communications a confirmation of the potentialities of the technique.

In the revised manuscript, we have introduced the previous report on the use of picosecond ultrasonics for the evaluation of deformation potentials by including a sentence and the two references proposed by the reviewer in the third paragraph.

2) The authors develop an analysis based on the measured amplitude of coherent acoustic phonon signals and relate (see SI) these values to the electron-hole-acoustic phonon deformation potential. In the present manuscript, there is the following statement

"We know that, for the same photoexcited carrier density, the difference in amplitude for different pump wavelengths is solely due to the change of energy of the pump photons »

This statement is not true in general. The authors have to support this approximation with more serious arguments. As the authors certainly know, the amplitude of the CAP does depend on the detection and generation processes. The detection is performed at the Brillouin frequency. So the amplitude of the photoinduced strain does not depend only on the values of the deformation potential, the thermal expansion, the bulk modulus, heat capacity. The detected amplitude does depend on the amplitude of the photoinduced strain spectrum at the Brillouin frequency. It is established for more than 20 years that the photoinduced strain (in the simplest form) has to be written as a function of the acoustic phonons frequency ω and of the physical parameters of generation (see well established theory and experimental confirmations in Gusev & Karabutov Laser optoacoustics, AIP, New York 1995. Wright et al, Phys. Rev. 64, 081202 (2001), Babilotte et al Phys. Rev. B 81, 245207 (2010), Young et al Phys. Rev. B, 86, 155207 (2012)):

As a example that the authors know, for a bipolar strain pulse (typically the one the authors might generate in their experiments), this spectrum $S(\omega)$ can be written as :

$$S(\omega) \propto \omega \tau / (1 + \omega^2 \tau^2)$$

where τ is a characteristic time that can be associated to different physical parameter during the generation process. In the case the spatial extension of the pulse strain is only limited by the skin depth of pump penetration (that depends on the pump wavelength), τ corresponds to the time propagation of the acoustic phonon within the skin of pump light. And τ is a function of the pump wavelength in general. This spectrum expression can be developed also in the case of diffusion of hot carriers and of heat, see references above.

- The authors have to correct first this wrong statement.

- Secondly, the consequence of this is that, without taking into account the photogenerated acoustic pulse spectrum, the authors cannot apply their model straightforwardly as they do. As a consequence, they cannot claim it is straightforward to evaluate the deformation potential parameter and the carriers mobility. The authors should present/develop a calculation of the photoinduced strain taking into account (at least) the effect of the pump optical absorption coefficient (which depends on the pump wavelength). The authors know certainly the optical properties of this perovskite.

Maybe this rapid calculation will show that their approximation appears to be right, but they do not show in the current manuscript that all the necessary checks have been taken. The readers have to be convinced first.

The authors provide in the SI a calculation about GaAs that they consider as a proof of their method. I do not believe it. GaAs has been studied by many many authors (Japan, France, UK, US, Taiwan, etc...) and all know, like the authors, that the amplitude at the Brillouin frequency does not depend only on the deformation potential (see Gusev & Karabutov Laser optoacoustics, AIP, New York 1995. Wright et al, Phys. Rev. 64, 081202 (2001), Babilotte et al Phys. Rev. B 81, 245207 (2010), Young et al Phys. Rev. B, 86, 155207 (2012)). I think the example of GaAs is not convincing.

I think that the oversimplified model used by the authors may explain why they observe a so large discrepancy in the carriers mobility with other tabulated data (see comparison in the Table 1).

- In this novel manuscript, we have followed the advices from the reviewer. First, we have modified the sentence he is referring to in the supplementary information, and added a few sentences in the manuscript to explain that we are only probing a part of the whole strain at the Brillouin frequency and that the amplitude of this specific frequency component may differ as a function of the pump wavelength.
- We then performed the calculation proposed by the reviewer. We introduced a correction factor that take into account the difference in generation for different pump wavelength. Additionally, we took into consideration the effect of charge diffusion, which can be ignored in the case of $\text{CH}_3\text{NH}_3\text{PbI}_3$, but not for GaAs, and we thus separately calculate the correction factor for the thermoelastic and deformation potential components of the strain.

Thanks to the method proposed by the reviewer, we obtain better estimates of the deformation potentials, and thus of the carrier mobility. However, we do not believe that this is the reason for the discrepancy between our measured mobilities, which are taking into account only the scattering by acoustic phonons, and other experimental mobilities, which are sensitive to all scattering mechanisms.

3) The experimental Brillouin signals the authors report has a unusual time-dependence variation, in a sense that the Brillouin amplitude exhibits a kind of beating (appearance of a maximum in its amplitude at a time delay 100-150ps). The authors do not comment this effect. How can they fit then the Brillouin magnitude and extract the maximum of amplitude without explaining the beating effect ? This is impossible to my best knowledge.

I think the authors should take care of not concluding too rapidly without discussing at least different others possible effect (known in the literature). I mean that this kind of beating could have different origin :

- optical anisotropy of the perosvkite which could lead to the detection of different component of Brillouin frequencies (ordinary, extraordinary), and this in particular in the tetragonal phase $\langle 100 \rangle$ propagation direction (see such phenomena in another perosvkite BFO Lejman et al Nature Comm 7, 12345 (2016))
- inhomogeneities in the crystal that could lead to non-monotonous Brillouin signals (see A. Steigerwald, Appl. Phys. Lett. 94, 111910 (2009)).

The "unusual" time dependence noted by the reviewer is in fact an artefact due to the non-exponential decay of the transient reflectivity. In our former analysis, we extracted the coherent component of the signal by subtracting multiple exponential decays over the whole signal. A consequence of this procedure was the subtraction of part of the early signal on the steepest part of the decay. This effect was only observed on the first period of the signal. Furthermore, this effect was observed for both for $\text{CH}_3\text{NH}_3\text{PbI}_3$ and GaAs, we can thus conclude that it was due to the analysis. We now use a different extraction procedure and obtain a clean damped sine as expected for each signals.

4) Finally, it is difficult to follow the discussion about the trap dynamics and it appears somehow speculative at this stage. Why choosing the time 10ps to and to deduce so fundamentals properties like traps energy ??

In the revised manuscript, we added a sentence and a reference to justify the choice of 10 ps delay to study the traps. In order to extract the traps energy, we need to observe carrier in the trap states. The characteristic carrier trapping time in $\text{CH}_3\text{NH}_3\text{PbI}_3$, is on the order of 10 ps, while the recombination of

trap states happens on the time scale around 100 ps. By probing at 10 ps, we can thus observe a population in those trap states.

Concerning the reason why we want to extract the value of the traps energy, it can be seen in Eq. S6. Indeed, the difference in the thermoelastic contribution of the two samples is directly related to the density of traps and trap energy. We thus determined the trap energy to calculate the trap density, which enables the estimation of the deformation potential.

Reviewer #2 (Remarks to the Author):

1. The major problem concerns how the experimental results are presented. The author stated in the Abstract and a few other places that "...measure the acoustic phonon-limited electron and hole mobility using coherent acoustic phonons generated by femtosecond laser pulses". However, the mobility is not measured directly in this work, and the central quantities of the experiment should be the deformation potentials that are subsequently used to estimate the charge carrier mobility. I believe that measurement of the deformation potentials is already an important result, and the authors need to revise related parts of the paper to focus on what is really studied in the experiments.

In the revised manuscript, we have tried to follow the reviewer's advices. For instance, we do not state anymore that we are measuring the mobility limit, but that we determine the deformation potentials which give an insight in electron phonon scattering and thus mobility.

2. In this work, the pump-probe measurement is carefully designed to obtain the deformation potentials. Even though, some important details and/or discussions are not included in the paper. For example, are all calculations involving the $(\Delta R)/R$ amplitudes compensated for the different absorption coefficients at different wavelengths? (a figure of the absorption spectra in the supplementary may also help) As a result of different absorption coefficients, do they have different penetration depths? And will this affect the amplitude of CAP signals?

As mention in response to reviewer #1, in this novel version of the manuscript, we have performed our calculations taking into account the effect of penetration depth of the pump and carrier diffusion on the frequency content of the generated acoustic strain.

3. On how the impurity densities are determined. (1) It seems that, the authors measure the impurity scattering of phonons and assume that scattering of charge carriers is proportional to that of phonons. Is this an assumption or can it be justified? If it is an assumption, it should be stated explicitly in the paper. (2) Are the absorption spectra the same for the two samples annealed at different temperatures? If not, will this also affect the time decay of the signals in Fig. 4? (It is claimed in Ref. 14 that, "the attenuation of the reflectivity oscillations is principally due to the decay of the sensitivity function.", thus the decay should also depend on the penetration depth.)

(1) We have added some sentences in the manuscript, to show that the changes in the scattering of phonons is due to the creation of deficiencies in the crystal structure. We then justify the proportionality between phonon and charge scattering by citing a reference that demonstrate the role played by these specific vacancies on trapping.

(2) In order to avoid parasitic contributions to the decay of the signal, we have performed our experiments with a probe wavelength below the bandgap. In that case, probe absorption is only caused by trap absorption, which have a low absorption cross section. To clarify this point, we added a sentence in the manuscript, and a reference for the low absorption cross section.

4. How accurate are the deformation potentials obtained from the experiment? And how are the error bars in the electron and hole mobilities estimated in Table I?

In the manuscript, we have now added the error bars on the deformation potentials. The error bars are coming from the uncertainty on the scaling factor α of Eq. S2.

Reviewer #3 (Remarks to the Author):

First, studies of the temperature-dependence of the mobility in MAPbI₃ has generally found a $T^{-3/2}$ dependence (J. Phys. Chem. Lett. 2015, 6 (24), 4991–4996.). This looks like a mobility limited by acoustic phonon scattering, consistent with the author's view. The problem is that the mobility measured in these same experiments is orders of magnitude lower than that calculated by the present authors. If the mobility were limited by defect scattering in real samples (as these authors claim) mobility should go UP with increasing temperature, not down in the experiment. They must address this disconnect.

Second, the assertion that their calculated mobility represents the intrinsic upper limit assumes that there are no other intrinsic mechanisms that could further limit the mobility. This might be true in a non-polar semiconductor, however the strongly polar, even ionic, nature of MAPbI₃ lattice suggests that the Frölich interaction should be important, leading to carrier scattering that is dominated by scattering from optical phonons as is observed in other polar semiconductors like GaAs. A recent paper by the Herz group argues strongly that this is the case. (DOI: 10.1038/ncomms11755).

In the revised manuscript, we have importantly modified the second paragraph to place our work in the current existing literature. We more specifically highlighted what is the current understanding of the charge transport and what concepts still need to be clarified.

We also added a paragraph at the end of the manuscript to address the discrepancy between our results and the existing literature. In the literature, there is a wealth of results that may sometime seems contradicting, like the two references provided by the reviewer, in which one group demonstrated that mobility is limited by acoustic phonon and for the other, it is limited by optical phonons. Taking these and other results from literature into consideration, we have tried to develop an understanding of our results based on formation of a polaron due to coupling of charges with the strongly polar lattice of CH₃NH₃PbI₃.

Reviewers' Comments:

Reviewer #1 (Remarks to the Author)

The authors have improved significantly the manuscript by taking a bit more care regarding the analysis. Additional theoretical considerations provide a more convincing approach. However, the authors have to underline in the manuscript that their method remains an indirect method for carrier mobility evaluation. Moreover, I encourage the authors to be careful when they connect directly the attenuation of acoustic phonons to traps concentration. There is a relation but not so straightforward. So, it is necessary that the reader have this information clearly in mind, ie the approximation made by the authors.

Apart these last remarks, I think the paper is interesting now for a broad audience including communities of materials science for photovoltaics and solid state physics. I think the paper can be published in Nat Comm.

Reviewer #2 (Remarks to the Author)

The revised manuscript has addressed most of my previous concerns, especially on the presentation of the results, and the correction factors for different pump wavelengths. I do find a minor typo in table 1 though, where the second set of theoretical data should come from Ref. 17, rather than Ref. 13. The paper is publishable in Nature Comm. after it is corrected.

Reviewer #3 (Remarks to the Author)

With the much improved discussion of existing literature, I find the present manuscript to be valuable contribution that deserves immediate publication.

Point by point response to reviewers' comments

Reviewer 1:

In the modified manuscript, we have added a sentence in the third and last paragraphs to emphasize that the method we have developed doesn't directly provide the mobility.

We have also added a sentence to clarify the assumption that we are making concerning the estimation of the density of defect that the lifetime of phonons solely depends on the density of defects.

Reviewer 2:

We have modified the reference in table 1 from 13 to 17.